# Mechanistic Insights about Sorafenib-, Valproic Acid- and Metformin-Induced Cell Death in Hepatocellular Carcinoma

**DOI:** 10.3390/ijms25031760

**Published:** 2024-02-01

**Authors:** Edgar Xchel Franco-Juárez, Vianey González-Villasana, María Elena Camacho-Moll, Luisa Rendón-Garlant, Patricia Nefertari Ramírez-Flores, Beatriz Silva-Ramírez, Katia Peñuelas-Urquides, Ethel Daniela Cabello-Ruiz, Fabiola Castorena-Torres, Mario Bermúdez de León

**Affiliations:** 1Departamento de Biología Molecular, Centro de Investigación Biomédica del Noreste, Instituto Mexicano del Seguro Social, Monterrey 64720, Nuevo Leon, Mexico; xchelfj@yahoo.com (E.X.F.-J.); maria.camachomo@imss.gob.mx (M.E.C.-M.); a00835273@tec.mx (P.N.R.-F.); katia.penuelasu@imss.gob.mx (K.P.-U.); 2Facultad de Ciencias Biológicas, Universidad Autónoma de Nuevo León, San Nicolás de los Garza 66451, Nuevo Leon, Mexico; vianey.gonzalezvl@uanl.edu.mx (V.G.-V.); luisa.rendon29@gmail.com (L.R.-G.); ethel.cabellorz@uanl.edu.mx (E.D.C.-R.); 3Tecnológico de Monterrey, Escuela de Medicina y Ciencias de la Salud, Monterrey 64710, Nuevo Leon, Mexico; fcastorena@tec.mx; 4Departamento de Inmunogenética, Centro de Investigación Biomédica del Noreste, Instituto Mexicano del Seguro Social, Monterrey 64720, Nuevo Leon, Mexico; silbear2002@yahoo.es

**Keywords:** hepatocellular carcinoma, sorafenib, apoptosis, autophagy, metformin, valproic acid, cell death, drug repurposing

## Abstract

Hepatocellular carcinoma (HCC) is among the main causes of death by cancer worldwide, representing about 80–90% of all liver cancers. Treatments available for advanced HCC include atezolizumab, bevacizumab, sorafenib, among others. Atezolizumab and bevacizumab are immunological options recently incorporated into first-line treatments, along with sorafenib, for which great treatment achievements have been reached. However, sorafenib resistance is developed in most patients, and therapeutical combinations targeting cancer hallmark mechanisms and intracellular signaling have been proposed. In this review, we compiled evidence of the mechanisms of cell death caused by sorafenib administered alone or in combination with valproic acid and metformin and discussed them from a molecular perspective.

## 1. Introduction

The most recent report provided by GLOBOCAN in 2020 positions hepatocellular carcinoma (HCC) in the sixth place with respect to worldwide cancer incidence, with 905,677 new cases, and in the third place with respect to cancer-related mortality, with 830,180 deaths [1]. From 2008 to 2020, the rate of new HCC cases increased by ~20%, from 748,000 to 905,677, and HCC-related mortality showed the same trend, with an increase of ~19%, from 696,000 to 830,180 cases. The relative prevalence remained constant from 2008 to 2020 (Table 1). Of note, the impact of the COVID-19 pandemic on HCC incidence and mortality is still to be evaluated, as some reports indicated modifications in the management of early-stage patients [2].

The most common type of primary liver cancer is hepatocellular carcinoma (HCC), accounting for 80 to 90% of all patients diagnosed with liver cancer, followed by intrahepatic cholangiocarcinoma, mixed hepatocellular carcinoma/cholangiocarcinoma and others less frequently observed such as children hepatoblastoma [3,4,5].

The risk factors related to HCC development are mainly grouped in two categories: (1) biotic factors, which include chronic viral hepatitis B or C or aflatoxin exposure; (2) abiotic factors, such as alcohol consumption, tobacco smoking, obesity, diabetes and metabolic syndrome [6]. The incidence by risk factor varies depending on the geographic region. For example, chronic hepatitis B infection is the main risk factor for HCC in some regions of East Asia or sub-Saharan Africa, whereas in Europe, hepatitis C is the main risk factor [6,7]. Two common features preceding most HCC cases are chronic inflammation and cirrhosis [3]. Nonetheless, evidence of HCC development in the absence of a cirrhotic environment or non-alcoholic steatohepatitis has been described [8]. It has been suggested that patients with at least one major risk factor should be included in a surveillance program due to the absence of symptoms in early stages of the disease and the association between early tumor detection and improved survival [9].

According to the Barcelona Clinic Liver Cancer guidelines, HCC is classified in five stages depending on the size and number of tumor nodules, extrahepatic spread, cancer-related symptoms or patient performance status [10]. Patients with HCC in the 0 and A stages can receive surgical treatment such as tumor ablation, liver transplant and surgical resection and have an expected survival or >5 years, whereas the surgical treatment for patients with HCC in the B stage is liver transplant, with an expected survival of >5 years, or trans-arterial chemoembolization, with an expected survival of 2.5 years [10]. In some cases, stage B patients can receive a systemic treatment consisting of sorafenib or lenvatinib, when not eligible for immunotherapy [10]. Patients diagnosed with advanced hepatocarcinoma (stage C) are only eligible for systemic treatment [10]. The life expectancy of patients with end-stage (D) HCC receiving a systemic treatment and the best supportive care is >2 years and 3 months [10]. 

Immunotherapy has been developed over the last few years as an alternative treatment for different diseases including cancer. Its main components are monoclonal antibodies that target membrane proteins in immune system cells or tumor cells. For HCC, three main targets have been explored, which are programmed cell death protein-1 (PD-1), programmed cell death ligand-1 (PD-L1) and cytotoxic T-lymphocyte-associated protein 4 (CTLA4) [11,12,13]. Nivolumab is a PD-1 inhibitor usually used alongside with ipilimumab, a CTLA4 inhibitor [11]. Both drugs have been tested for the treatment of several types of cancer such as lung cancer, melanoma and Hodgkin’s lymphoma, among others [11,12,13]. A multicenter, open-labelled, multicohort, phase 1/2 study demonstrated the efficient antitumoral activity of nivolumab; however, no significant improvement was observed when compared with sorafenib [14]. Camrelizumab, another PD-1 inhibitor commonly used with rivoceranib, a tyrosine kinase inhibitor, showed an improvement in overall survival and progress-free survival compared to the single-drug treatment with sorafenib [15]. The treatment with tremelimumab, a CTLA-4 inhibitor, combined with durvalumab, an anti-PD-L1 antibody, demonstrated an increase in overall survival compared to the treatment with sorafenib in the clinical trial HIMALAYA [16]. A phase 1b trial demonstrated that in patients with unresectable hepatocellular carcinoma, treatment with atezolizumab, an anti-PD-L1 antibody, combined with bevacizumab, a vascular endothelial growth factor A (VEGF-A) inhibitor, resulted in better overall and progression-free survival compared to sorafenib administration [17]. This treatment has been already approved for patients with unresectable hepatocellular carcinoma [18,19]. Despite the promising results with combined treatments, sorafenib remains a first-line treatment since its approval in 2008 by the Food and Drug Administration (FDA) [20]. Sorafenib is an orally administered tyrosine kinase receptor inhibitor [21], which is prescribed in a dose of 400–800 mg/day and is considered the standard treatment for patients with stage B or C HCC who are not eligible for, or progress despite, locoregional therapies [22]. Patients treated with sorafenib showed an improvement in overall survival of ~3 months compared to those administered a placebo [23,24]. Unfortunately, most patients will develop sorafenib resistance, and an approach to overcome this issue could be targeting regulated cell death (RCD) by combining molecules that drive or facilitate the concomitant regulation of one or different cell death mechanisms. Sorafenib itself was shown to induce different RCD programs such as apoptosis [25,26], autophagy [27,28], ferroptosis [29,30], pyroptosis [31] and other cell death modalities such as accidental cell death [32,33]. In the present review, the mechanisms of cell death caused by sorafenib administered alone or in combination with valproic acid and metformin, two drugs that could be repurposed and that have been shown to increase sorafenib-induced cell death, are described. 

## 2. Cell Death Mechanisms Related to Sorafenib

According to the Nomenclature Committee on Cell Death, cell death can occur accidentally or be regulated [34]. In the former case, cell death, also referred to as necrosis, is caused by a plethora of physical or chemical signals and does not depend on any known cell death program [35]. Accidental cell death is undesirable, mainly due to the release of pro-inflammatory factors that ultimately promote tumor growth [36]. On the other hand, RCD is a type of death that occurs as a consequence of different stimuli and, unlike the accidental form, it relies on highly regulated mechanisms, including apoptosis, autophagy, ferroptosis, pyroptosis, necroptosis mechanisms and others [34]. To further explore sorafenib antitumoral properties, some studies evaluated RCD markers and suggested that sorafenib mainly triggers cell death mechanisms [25,26,27,28,29,30,37,38]. More recently, its role in other antitumor mechanisms such as pyroptosis in HCC-associated macrophages [31] and necroptosis [39] was suggested. 

Studies carried out on HCC cells showed a dose-dependent cytotoxic effect of sorafenib [40]. This was demonstrated by the increased rate of phosphatidyl serine (PS)-positive HCC cells [33,40,41] and was also shown in non-solid tumors [42]. Further characterization showed that the cell death mechanism activated by sorafenib could be apoptosis [21,25,43]. 

There are two forms of apoptosis, i.e., extrinsic and intrinsic. The extrinsic apoptotic pathway is activated by the interaction between cell surface death receptors of the tumor necrosis factor family and their respective ligands, such as the interactions between Fas cell surface death receptor (FAS) and Fas ligand (FAS-L), tumor necrosis factor receptor (TNFR) and and tumor necrosis factor alpha (TNF-α) and others, which regulate the activation and execution of apoptosis [44]. In a sequence of events, upon activation of a cell death receptor, signals are transduced mainly due to adaptor proteins like Fas-associated death domain (FADD) or tumor necrosis factor receptor type-1-associated death domain (TRADD) that assemble an autocatalytic complex with pro-caspase 8, which mediates its own cleavage, resulting in the activation of caspase 8 (cleaved caspase 8) and of executioner caspases 3–7 and culminating in cellular demolition [44,45]. 

The intrinsic form of apoptosis is triggered mainly by DNA damage, increased levels of intracellular ions and other cellular disturbances that induce B-cell lymphoma 2 protein family-regulated signals [34]. The balance between pro-apoptotic proteins of the BCL-2 family, which are BCL-2-associated X apoptosis regulator (BAX), BCL-2-associated agonist of cell death (BAD), p53-upregulated modulator of apoptosis (PUMA), BH3-interacting domain death agonist (BID), BCL-2-like 11 (BIM), and their counterparts with anti-apoptotic properties, such as BCL-2 apoptosis regulator (BCL-2), BCL-2-like 2 (BCL-W), BCL-2-like 1 (BCL-xL), MCL-1 apoptosis regulator (MCL-1) and BCL-2-related protein A1 (BCL-2A1), determines the cell apoptotic state [46,47,48].

Targeting BCL-2 proteins has been a successful strategy used in many FDA-approved cancer therapies and relies on the potential of these proteins to activate RCD [46]. This is the case for sorafenib, an antiproliferative drug which inhibits tyrosine kinase receptors, like vascular endothelial growth factor receptor, fibroblast growth factor receptor and other molecules, along with some components of the mitogen-activated protein kinase pathway such as RAF-1 proto-oncogene, serine threonine kinase, mitogen-activated protein kinase kinase 1 or mitogen-activated protein kinase 3 [21]. 

Cells exposed to sorafenib showed a pro-apoptotic gene expression profile. This was demonstrated in vitro by the change in the mRNA dynamics of BCL-2 family pro-apoptotic members like *BIM* and *PUMA* accompanied with a reduction in the mRNA levels of anti-apoptotic members like baculoviral IAP repeat-containing 5 (*SURVIVIN*) and *MCL-1* [25,26]. Further evaluations of critical points showed that sorafenib promoted cytosolic cytochrome C (Cyt C) release [43] and the activation of executioner caspases [43,49]. Features of apoptotic cell death were observed in HCC cells treated with sorafenib, like DNA fragmentation confirmed by the terminal deoxynucleotidyl transferase dUTP nick-end labeling (TUNEL) assay in vitro [21] and in vivo [41], the presence of phosphatidyl serine-positive cells and the cleavage of poly(ADP-ribose) polymerase (PARP) [33,40,41,50].

Studies conducted to elucidate the potential pathways involved in sorafenib-induced apoptosis showed that the tumor protein p53 (p53) and the E2F transcription factor 1 (E2F1) could be responsible for apoptosis activation and regulation [51,52]. p53 protein is a tumor suppressor with a relatively short life and is activated upon cellular stress like DNA damage, mitogenic chronic stimulation, genomic instability or hypoxia [53,54]. Upon activation, p53 participates in different cell processes like senescence, cell cycle arrest, DNA repair or apoptosis [55]. A p53 transcription-dependent apoptosis mechanism has been described in sorafenib-treated HCC cells [40]. In transcription-dependent apoptosis, p53 activation recruits the basal transcription machinery and different histone acetylases to different promoters of apoptosis-related genes such as phorbol-12-myristate-13-acetate-induced protein 1 (*NOXA*), apoptotic peptidase-activating factor 1 (*APAF1*), caspase 1, caspase 6, *FAS* and others such as *BAX* or *PUMA* [48,55]. Also, p53 can regulate the expression of other transcription factors such as forkhead box M1 (*FOXM1*), whose activity is upregulated in HCC progression [51,56]. 

Apoptosis also occurs in a p53 transcription-independent manner. A direct interaction between p53 and BCL-2 protein family members results in p53 acetylation (at K120, K381 and K382) following DNA damage, which promotes the transcription of apoptotic-related genes such as *BAX* or *PUMA* [53,57,58,59]. Garten and collaborators (2019) found that sorafenib increased the levels of p53 acetylation at lysine 382, and concomitantly, PUMA transcription was also increased [40]. (2019) found that sorafenib increased the levels of p53 acetylation at lysine 382, and concomitantly, PUMA transcription was also increased [40]. The exact mechanism involved in sorafenib-mediated apoptosis remains elusive, but evidence suggests that it could at least in part entail p53-independent transcriptional activity.

Another important transcription factor modified by sorafenib during apoptosis is E2F1, which belongs to the E2F family that comprises eight members (E2F1-8) [60]. The E2F family is involved in the regulation of different processes such as cell cycle progression and apoptosis, potentially modulated by sorafenib [52,61,62,63,64]. Information regarding the role of E2F1 in apoptosis was previously compiled [65].

The E2F1/p53 transcription-dependent mechanism relies on the expression of multiple genes that have been reviewed elsewhere [66], including those known to promote apoptosis such as *PUMA*, *NOXA* and *BIM* [67,68]. Some studies focusing on the role of E2F1 in HCC cells revealed that E2F1 mRNA and protein levels decreased in a time-dependent manner in HCC cell lines exposed to sorafenib [52]. 

Previous studies demonstrated that in HCC tissues, E2F1 has a nuclear and cytoplasmic expression, which is increased compared to that in non-cancerous tissues [61]. The regulation of E2F1 is directed by retinoblastoma (Rb) and pocket proteins [69] in vitro [70]. However, the p53– and Rb–E2F1-related putative activity of sorafenib involving the regulation of the mRNA and protein levels of Rb and E2F1 remains to be fully elucidated. Further studies are required to evaluate the activity of these proteins in a controlled environment and in the presence of different inhibitors to elucidate the involvement of p53- and E2F1-regulated cell death in sorafenib effects.

Autophagy is another form of RCD that was described upon sorafenib treatment [71]. Furthermore, it was shown that the cellular sensitivity to sorafenib can be affected by the modification of autophagic proteins, such as the conversion of microtubule-associated proteins light-chain B-I (LC3B-I) to microtubule-associated proteins light-chain B-II (LC3B-II) [72]. In HCC, it was shown that the molecular mechanisms by which sorafenib induces autophagy could be independent of AMP-activated protein kinase (AMPK) and involve a reduction in MCL-1 levels, which in turns would inactivate signal transducer and activator of transcription 3 factor (STAT3), causing the accumulation of beclin-1 (BECN-1) and other proteins indicating autophagosome formation, such as LC3B-II accumulation, as well as sequestosome 1 (p62) reduction [25,28,41,49,73]. Several studies reported a dose-dependent cytotoxic effect of sorafenib, along with the induction of the assembly of the autophagolysosome, which was accompanied by reduced cell viability and an increase in annexinV/Pi-positive cells in vitro, as well as by an increase in LC3-II conversion and autophagosome formation in tumor samples and a reduced tumor burden in HCC-bearing mice [37,41,74]. 

To our better knowledge, in the clinical setting, there is no evidence of an association between autophagy markers and sorafenib treatment. However, some studies evaluated the presence of autophagy-related proteins in patient HCC samples and found increased levels of LC3-II and a correlation with a longer overall survival [75,76]. 

Taking these data together, sorafenib can induce cell death via apoptosis and autophagy. Further investigations are required to elucidate its potential use as a modulator of the link between autophagy and apoptosis dependent on BECN1 and BCL-2. 

Ferroptosis is a novel form of RCD, recently reviewed in HCC [77]. The process of ferroptosis is dependent on the accumulation of iron and reactive oxygen species (ROS) [38]. It was previously reported that sorafenib-induced cell death occurs through an iron-dependent mechanism and that in HCC cells treated with sorafenib and the iron chelator deferoxamine (DFX), a reduction in the cytotoxic effects of sorafenib was observed, suggesting a DFX protective effect [29]. However, DFX does not prevent sorafenib from interacting with intracellular kinases, implying that sorafenib induces a cytotoxic effect that resembles ferroptosis [29]. Another mechanism that was associated with ferroptosis and sorafenib is system xc^−^ [30]. This system comprises solute carrier family 7 member 11 (SLC7A11) and solute carrier family 3 member 2 (SLC3A2). In a non-small cell lung carcinoma cell line, the upregulation of p53 inhibited SLC7A11 expression, therefore predisposing cells to ferroptosis [78,79]. The gene profiling of sorafenib-induced ferroptosis showed that NAD(P)H quinone dehydrogenase 1, hemeoxigenase 1, ferritin heavy chain 1 and metallothionein 1G were involved, and the knockdown of these genes and p62 sensitized HCC to ferroptosis upon sorafenib and erastin treatment [77].

Recently, it was shown that Src homology region 2 domain-containing phosphatase-1/STAT3 signaling axis-regulated coupling between BECN1 and SLC7A11 contributes to sorafenib-induced ferroptosis in HCC [80]. 

Controversial results were reported regarding sorafenib induction of ferroptosis, as other studies showed in several cell lines including HepG2 and HuH-7 that sorafenib by itself failed to trigger ferroptosis [81]. 

Pyroptosis is another type of RCD, which is mediated by the gasdermin family [82]. It was shown that sorafenib induced pyroptosis in macrophages and triggered natural killer cell-mediated cytotoxicity in hepatocellular carcinoma [31]. Specifically, Hage et al. detected caspase 1 in sorafenib-treated macrophages, which was indicative of the induction of pyroptosis. They also demonstrated that because of pyroptosis, cytotoxic natural killer cells became activated, leading to tumor cell death [31]. In other cancers, such as papillary thyroid carcinoma, sorafenib was shown to increase the expression of angiopoietin-like 7, cyclin-dependent kinase inhibitor 2A, dipeptidyl peptidase 4, dopamine receptor D4, iron–sulfur cluster assembly enzyme, phosphogluconate dehydrogenase, sulfiredoxin 1, transferrin, transferrin receptor and thioredoxin reductase 1 [83]. 

Necrosis causes irreversible cell injury, which culminates in cell death [84]. It was shown that in HCC cells, sorafenib increased necrosis, as demonstrated by the elevated counts of annexinV-negative/Pi-positive cells after 48 h of sorafenib treatment [27,41,85]. A summary of the mechanisms of sorafenib-induced cell death can be observed in Figure 1.

## 3. Cell Death Mechanisms Related to Valproic Acid

Valproic acid (VPA) has been recently repurposed and could be a potential candidate to facilitate RCD in HCC. Studies on the histone deacetylase inhibition activity of VPA showed changes in chromatin structure [86], which potentially modify gene expression. In HCC, an aberrant epigenetic profile was documented and associated with different events such as genomic instability, oncogenic signaling activation, modification in signaling regulation, impairment of DNA repairing systems and dysregulation of apoptotic gene expression [87].

In terms of cell death, VPA-induced RCD was evidenced by the induction of intrinsic apoptosis markers such as *BAX*, BCL2 antagonist/killer 1 (*BAK*) and *BIM* and a reduction in antiapoptotic members of the BCL-2 family [86,88]. VPA was also shown to activate the intrinsic apoptosis pathway in HCC through the induction of caspases 3 and 9 [89]. Extrinsic apoptosis was also shown to be induced by VPA, as in non-solid tumors, the cleavage of BH3-interacting domain death agonist (BID) by caspase 8 was observed [86], which is an event that facilitates the release of Cyt C from the mitochondria [90,91]. In nine leukemia cell lines, VPA induced cell death, as indicated by the observation of apoptotic changes such as DNA fragmentation, phosphatidyl serine externalization, Cyt C release from the mitochondria and activation of caspases 3, 8, and 9 [92]. In xenograft models, VPA was shown to induce apoptosis, as demonstrated by the TUNEL assay and the detection of cleaved caspase 3-positive cells, with a significant reduction of 36.4% in the mean tumor volume in the VPA-treated group compared the control group after 4 weeks of treatment (*p* < 0.01) [93]. To our knowledge, there is no information about apoptotic RCD markers associated with VPA treatment in the clinical setting.

Autophagy was also reported in pancreatic cancer after VPA treatment, as indicated by the increase in LC3B-II levels in the presence of chloroquine, an autophagy inhibitor, which was indicative of a complete autophagic flux [94]. Autophagy was also reported in metastatic thyroid cancer cells treated with VPA, as evidenced by the induction of LC3B-II and p62 [95]. A summary of the molecules involved in VPA-induced apoptosis and autophagy is shown in Figure 2. 

A literature search on ferroptosis and VPA in PubMed did not show any results. However, given that BCL-2, BID and SURVIVIN are involved in VPA-induced cell death, it is possible that ferroptosis occurs via p53 or STAT3, given that BCL-2 and BIM are downstream of p53, and SURVIVIN is downstream of STAT3. p53 and its role in ferroptosis were recently reviewed [96]. 

## 4. Cell Death Mechanisms Related to Metformin

Evans et al. (2005) suggested that the duration and dosage of metformin administration could reduce the risk of cancer in patients with diabetes [97]. A recent review summarized several meta-analysis and information related to metformin in HCC progression and metastasis and the potential signaling pathways modified by metformin, including those related to energy metabolism [98]. Metformin was associated with a reduced risk of HCC development, and its use correlated with better outcomes in different patient populations diagnosed with diabetes type 2, except in one retrospective study that reported a risk of 1.44 for HCC mortality in 212 metformin users [99]. However, in this same study, the authors found that the 5-year survival rate in the Malaysian population was lower compared to that in other Asian populations [99], where HCC is one of the main health issues due to its incidence and mortality [100].

Metformin repurposing has become of interest, given several reports demonstrating its potential as an anticancer drug in several oncogenic models via a mechanism involving cyclin-dependent kinase inhibitor 1A (p21), cleaved caspase 3, cleaved caspase 9 and inflammatory markers such as prostaglandin-endoperoxide synthase 2 (Cox-2), nuclear factor kappa B (NF-kB) and VEGF-A [101,102]. Metformin effects on cell viability were demonstrated to be dose-dependent, with different response profiles across diverse in vitro HCC models and the increase in annexinV/Pi-positive HCC cells [103,104,105]. In normal liver cells, metformin has limited effects on cell viability [106]. Studies on metformin apoptotic molecular mechanisms showed a significant reduction in *BCL-2* mRNA but not in *p53* or *BAX* [104] mRNA. Also, the induction of p53, BAX, cleaved PARP and cleaved caspase 3 along with a reduction in BCL-2 in HCC cells after metformin treatment was reported [103]. This proapoptotic profile induced by metformin was also observed in bladder cancer [107] and in osteosarcoma [108]. It was previously observed that metformin can reduce SURVIVIN levels through AMPK and the mechanistic target of rapamycin kinase (mTOR) axis [109]. A role for AKT serine/threonine kinase 1 (AKT) and mTOR was described in tumor samples from endometrial cancer patients and in gastric cancer cells treated with metformin [110,111]. 

Metformin is also capable of inducing other types of cell death such as autophagy, as it was shown that in HCC in vitro and in vivo models, the AMPK–mTOR axis was altered after metformin treatment [41]. The AMPK–mTOR pathway has been suggested as one of the cell energy sensor systems and is associated with the activation of the autophagic machinery [112,113]. AMPK is activated under energy starvation, which is sensed based on the intracellular ratio of ADP/ATP [98]. It was observed that low levels of activated AMPK are correlated with HCC occurrence [114]. Another study [111] reported that metformin failed to trigger the autophagic flux in two HCC lines and that the type of metformin-induced cell death was different in the two cell lines, mainly due to differences in the basal levels of autophagy potentially related to the activation status of AKT, an upstream regulator of the mTOR complex [115].

It was demonstrated that metformin can cause cell death by different mechanisms such as apoptosis and autophagy. Apoptosis was demonstrated in a human osteosarcoma cell line where cell shrinkage, condensation of chromatin and fragmentation of the nuclei were observed, together with an increased percentage of apoptotic cells, determined using Hoechst staining in flow cytometry experiments [108]. In this same study, the effects of metformin on osteosarcoma xenografts were investigated, and a remarkable reduction in tumor growth was observed; apoptosis was confirmed by the TUNEL assay and the presence of cleaved caspase and phosphorylated mitogen-activated protein kinase 8 [108].

Autophagy has been shown following metformin treatment. For instance in rat models of HCC, it was shown that metformin exerted antitumor effects via the AMPK-dependent pathway, demonstrating its potential use in the early stages of HCC development in rats [116]. This pathway was also reported in other studies, where autophagy occurred after metformin treatment, specifically, BECN1-independent autophagy through the AMPK–mTOR signaling pathway [111].

Deeper analyses of autophagic events demonstrated that metformin could enhance LC3B mRNA induction and increase the levels of BECN1, GABA type A receptor-associated protein like 2 and LC3-II proteins, accompanied by a reduction in p62 [117] and an increased signal related to LC3B puncta formation [41,112,118,119] and autophagic vacuoles in different HCC cell lines [41,115] and in gastric cells [111]. Metformin by itself can promote ferroptosis and increase sorafenib sensitivity in HCC by the downregulation of transcription factor 4 (ATF-4), which in turn inhibits the nuclear translocation of phosphorylated STAT3, promoting ferroptosis [120]. Metformin-induced ferroptosis was also reviewed in relation to diabetic retinopathy, aging, and other cancers [121]. In urological malignancies, mTOR was also included in a ferroptosis suppression mechanism [122].

Metformin was also shown to play a role in pyroptosis. For instance, it was shown to induce pyroptosis in leptin receptor-defective hepatocytes via overactivation of the AMPK axis [123], which in turn caused the accumulation of cleaved caspase 1, 11, 5, interleukin 18 (IL-18) and interleukin 1B (IL-1B). Furthermore, it was shown that metformin can inhibit HCC development through the induction of apoptosis and pyroptosis via forkhead box O3 [105].

Based on previous evidence, the role of metformin in RCD remains to be fully established, as insights into the molecular machinery it activates demonstrated that metformin could regulate apoptotic and autophagic cell death mechanisms, as well as be involved in pyroptosis and ferroptosis.

A summary of the molecules involved in metformin-induced cell death mechanisms is reported in Figure 3. 

## 5. Cell Death Mechanism Activated by the Combination of Valproic Acid and Sorafenib in HCC

In the history of disease, the administration of drug combinations has demonstrated strong and prolonged responses, a reduction in the doses required to achieve clinical outcomes, the simultaneous targeting of different pathways and the abrogation of drug resistance, in the treatment not only of infections but also of neoplasia [124]. Therefore, drug combinations could increase the benefits provided by single medications. On the other hand, drug repurposing in clinical research is expanding due to its advantages, as it allows treatment cost reductions and increases the probability of reaching the pharmacovigilance phase. Some of its main disadvantages are formulation-related difficulties, the establishment of clinically relevant concentrations, and the occurrence of drug–drug interactions [125]. The association of drug combination and drug repurposing could integrate the best advantages of both, while the disadvantages should be evaluated from a cost–benefit perspective.

In HCC cells, it was demonstrated that the combination of sorafenib with VPA synergistically inhibited HCC cell viability through the induction of intrinsic apoptosis involving p21, BAX, cleaved caspase 9, cleaved caspase 3 and cleaved PARP and the down-regulation of BCL-xL and SURVIVIN [33,126]. Furthermore, in a xenograft mouse model of HCC, the tumor burden decreased more effectively when a combination of VPA and sorafenib (rather than sorafenib alone) was administered [33]. Zhu (2017) proposed the involvement of a crosstalk between AKT and Notch receptor 3 (NOTCH3), with sorafenib increasing the levels of NOTCH3 and phosphorylated AKT, and VPA decreasing both of them [33]. This combination could overcome sorafenib resistance in HCC cells, as demonstrated in sorafenib-resistant HepG2 cells, where cell death was substantially increased by the combined treatment of VPA and sorafenib [127].

In vivo evaluations in a xenograft mouse model of HCC showed that compared to VPA, sorafenib further reduced tumor weight and volume; nonetheless, the combined treatment significantly reduced both parameters, compared to the single-drug treatment [33]. Western blot performed on tumor lysates showed that the combined-treatment group had higher levels of cleaved caspase 3 and 9 and cleaved PARP compared to the single-drug treatment groups [33].

To our knowledge, currently there is no clinical trial on advanced HCC evaluating sorafenib in combination with VPA. A summary of the molecules involved in sorafenib- and valproic acid-induced cell death mechanisms is reported in Figure 4.

## 6. Cell Death Mechanism Activated by the Combination of Metformin and Sorafenib in HCC

Another combinatorial approach involves sorafenib and metformin and was shown to cause a pronounced reduction in the viability of diverse HCC cell lines [128,129] and two human HCC-derived cell lines [130]. A reduction in tumor burden was reported in vivo following the co-administration of sorafenib and metformin, which was greater compared to those observed after single-drug treatments [41,129,131,132]. A time-dependent reduction in cell viability associated with a significant increase in pro-apoptotic proteins such as cleaved PARP and cleaved caspase 3 was observed in vitro and in vivo following the co-administration of metformin and sorafenib [41]. This was also confirmed by the presence of cleaved caspase 3 and TUNEL-positive cells and caused strong antitumoral effects potentially by the activation of the autophagic flux, as shown by the conversion of LC3B-I to LC3B-II, accompanied by a reduction in p62 mediated by the AKT–mTOR pathway [41]. 

The combined treatment increased lipid peroxidation and other markers related to ferroptosis, suggesting that metformin can facilitate this type of cell death and also overcome sorafenib-induced resistance via ATF4/STAT3 [120]. According to the ClinicalTrials.gov database [133], up to December 2023, only one clinical trial is evaluating the effects of sorafenib and sorafenib/metformin in HCC patients, but no results have been posted yet (study ID: NCT02672488). Ferroptosis can also be induced via the p62–Kelch-like ECH-associated protein 1–NF-E2-related factor 2 pathway after the combined treatment with sorafenib and metformin [134].

A list of markers associated with pyroptosis in HCC was published, which includes BAK, BAX, caspase 1, 3 and 5, gasdermin A and E, high-mobility group box 1, interleukin 18, 1A and 1B, p53, tumor protein 63, among others [135,136]. Of importance, among these genes, p53 is of interest, given its involvement in metformin- and sorafenib-related mechanisms of cell death. It was shown that the analysis of the clusters of pyroptosis is useful to predict prognosis in patients with HCC [135].

A summary of the molecules involved in cell death induced by the sorafenib and metformin combination treatment is reported in Figure 5.

## 7. Future Directions and Challenges

As shown above, evidence indicates that the cell viability and tumor burden reductions induced by sorafenib can be enhanced by VPA and metformin, probably due to an increase in signal transduction by the apoptotic and autophagic machinery; it also suggests that the administration of these combinations and the presence of other diseases such as diabetes should be further evaluated to identify the limitations of this combinatorial approach.

The present study provides evidence to support efficacy and safety studies on these combinations that should be carried out in animal models to inform clinical trials, given that although the combined treatment of metformin and sorafenib in diabetic patients was not so successful, this could not be the case for other malignancies such as hepatocellular carcinoma. Sorafenib resistance will remain a challenge until new therapies are developed, some of which might involve, as we propose, drug combinations and repurposed drugs, which could target several pathways at the same time and might exert synergistical effects, thus greatly benefiting HCC patients. Further elucidation of the mechanisms of cell death caused by the molecules discussed in the present study should be carried out to develop new targeted therapies.

## 8. Limitations

Further investigation is needed to better understand the role of drug-induced RCD and the regulatory mechanisms involved in an oncogenic environment, considering the presence of other diseases in patients. The available clinical evidence is not sufficient, as the studies conducted so far examined small samples; therefore, it is difficult to establish differences in sorafenib responses in different populations. This research only included studies from the PubMed library in English language. 

## 9. Conclusions

VPA and metformin could enhance sorafenib-induced cell death mainly through an increase in the levels of different regulatory and executioner proteins involved in different cell death mechanisms. Putative interaction nodes are shared between apoptosis and autophagy intracellular signaling pathways but also with intracellular signaling pathways affected by the combination of sorafenib with VPA or metformin. Also, metformin participates in the three main cell death mechanisms triggered by sorafenib, suggesting that by itself is capable of modulating different pathways and that perhaps its sensitizing effects can be related to its concentration used in treatment.

## Figures and Tables

**Figure 1 ijms-25-01760-f001:**
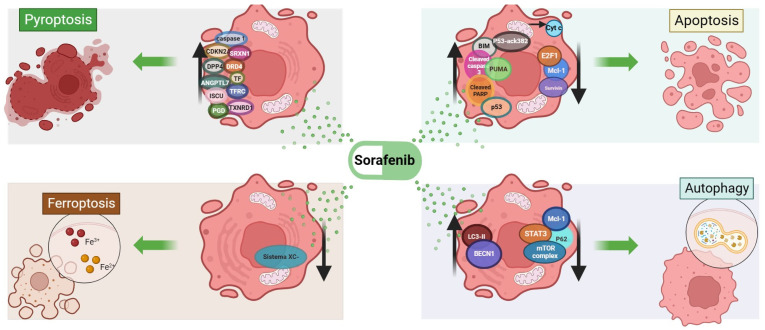
Mechanisms of sorafenib-induced pyroptosis, ferroptosis, apoptosis and autophagy. ANGPTL7, angiopoietin-like 7; BIM, BCL-2-like 11; BECN1, beclin-1; Cyt C, cytochrome C; CDKN2A, cyclin-dependent kinase inhibitor 2A; DPP4, dipeptidyl peptidase 4; DRD4, dopamine receptor D4; FOXM1, forkhead box M1; E2F1, E2F transcription factor 1; ISCU, iron–sulfur cluster assembly enzyme; LC3B-II, microtubule-associated proteins light chain B-II; MCL-1, MCL-1 apoptosis regulator; p53, tumor protein p53; p53-acK382, tumor protein p53 acetylated at lysine 382; p62, sequestosome 1; PARP, poly(ADP-ribose) polymerase 1; PGD, phosphogluconate dehydrogenase; PUMA, p53-upregulated modulator of apoptosis; SRXN1, sulfiredoxin 1; STAT3, signal transducer and activator of transcription 3; SURVIVIN, baculoviral IAP repeat-containing 5; TF, transferrin; TFRC, transferrin receptor; TXNRD1, thioredoxin reductase 1. Image created in BioRender.com (accessed on 11 December 2023).

**Figure 2 ijms-25-01760-f002:**
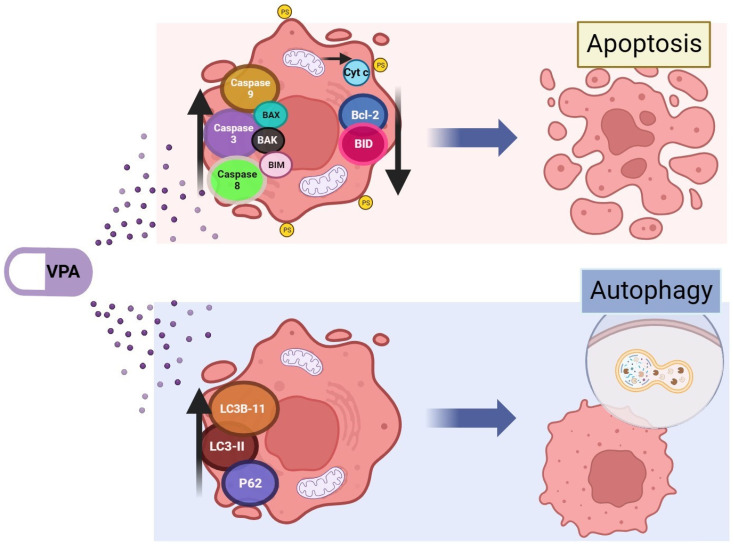
Apoptosis- and autophagy-related proteins modified by valproic acid. Valproic acid (VPA) induces the accumulation of different proteins associated with an apoptotic profile in HCC. BAX, BCL-2-associated X apoptosis regulator; BAK, BCL2 antagonist/killer 1; BIM, BCL-2-like protein 11; BCL-2, BCL-2 apoptosis regulator; BID, BH3-interacting domain death agonist; Cyt c, dytochrome C; LC3B-I and -II, microtubule-associated proteins light chain B-I and -II; p62, sequestosome 1. Image created in BioRender.com (accessed on 11 December 2023).

**Figure 3 ijms-25-01760-f003:**
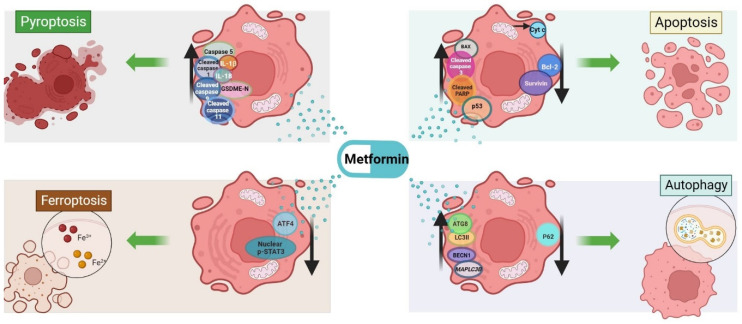
Metformin induces apoptotic and autophagic profiles in HCC cells. ATF-4, activating transcription factor 4; AMPK, AMP-activated protein kinase; ATG8, GABA type A receptor-associated protein like 2; BAX, BCL-2-associated X, apoptosis regulator; BCL-2, BCL-2 apoptosis regulator; BECN1, beclin-1; Cyt c, cytochrome C; FOXO3, forkhead box O3; GSDME, gasdermin E; IL-18, interleukin 18; IL-1B, interleukin 1B; LC3B-II, microtubule-associated proteins light chain B-II; mTOR, mechanistic target of rapamycin kinase; p53, tumor protein p53; p62, sequestosome 1; PARP, poly(ADP-ribose) polymerase 1; p-STAT3, phosphorylated signal transducer and activator of transcription 3; SURVIVIN, baculoviral IAP repeat-containing 5. Image created in BioRender.com (accessed on 11 December 2023).

**Figure 4 ijms-25-01760-f004:**
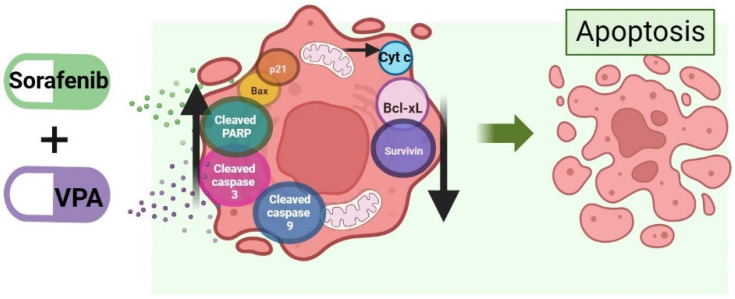
Apoptosis induced by sorafenib combined with valproic acid or metformin. AKT, AKT serine/threonine kinase 1; BAX, BCL-2-associated X; apoptosis regulator; BCL-XL, BCL-2-like 1; Cyt c, cytochrome C; NOTCH3, Notch receptor 3; p21, cyclin dependent kinase inhibitor 1A; PARP, poly(ADP-ribose) polymerase 1; SURVIVIN, baculoviral IAP repeat-containing 5. Image created in BioRender.com (accessed on 11 December 2023).

**Figure 5 ijms-25-01760-f005:**
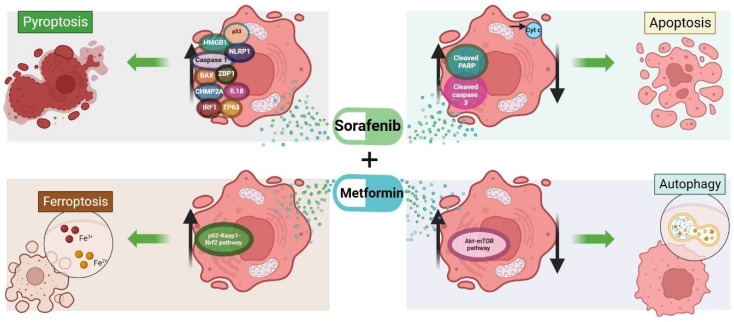
Cell death mechanisms induced by sorafenib combined with metformin in HCC. AKT, AKT serine/threonine kinase 1; BAX, BCL-2-associated X, apoptosis regulator; GSDMA, gasdermin A; GSDME, gasdermin E; HMGB1, high-mobility group box 1; IL-1A, interleukin 1A; Il-1B, interleukin 1B; IL-18, interleukin 18; Keap1, Kelch-like ECH-associated protein 1; LC3B-I and -II, microtubule-associated proteins light chain B-I and -II; Nrf2, NF-E2-related factor 2; p53, tumor protein p53; p62, sequestosome 1; PARP, poly(ADP-ribose) polymerase 1; mTOR, mechanistic target of rapamycin kinase; p63, tumor protein p63; ZBP1, Z-DNA-binding protein 1. Image created in BioRender.com (accessed on 11 December 2023).

**Table 1 ijms-25-01760-t001:** Worldwide liver cancer incidence and mortality from 2008 to 2020.

Year	Incidence	Rank	Mortality	Rank	World Population	% of Relative Prevalence
2008	748,000 ^1^	6	696,000 ^1^	3	6,789,088,686 ^2^	0.011
2012	782,000 ^1^	6	745,000 ^1^	2	7,125,828,059 ^2^	0.010
2018	841,080 ^1^	6	781,631 ^1^	3	7,631,091,040 ^2^	0.011
2020	905,677 ^1^	6	830,180 ^1^	4	7,794,798,739 ^2^	0.011

^1^ Incidence and mortality data were retrieved from GLOBOCAN (https://gco.iarc.fr/ (accessed on 23 September 2023), and ^2^ estimated world population from Worldometer in September 2023 (https://www.worldometers.info/ (accessed on 23 September 2023). Percentage of relative prevalence was determined as year incidence/year world population.

## Data Availability

Data from the manuscripts reviewed in the present study are available under reasonable request.

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
