# Peer review of "Mechanistic Insights about Sorafenib-, Valproic Acid- and Metformin-Induced Cell Death in Hepatocellular Carcinoma"

_ijms, 2024, doi:10.3390/ijms25031760_

Round 1
Reviewer 1 Report
Comments and Suggestions for Authors
The manuscript by Edgar X. Franco-Juárez et al entitled “Mechanistic findings of cell death in drug combinations with sorafenib against liver cancer: from basic research to clinical implementation” is a review manuscript in which authors attempt to provide information on the potential combination of this b-raf inhibitor and other drugs to treat liver cancer. Despite the fact that the subject matter is interesting, this reviewer finds it very difficult to understand the take-home message of this review:
The drug combinations implied by the authors in the title are in fact combinations with VPA and metformin only. However, even this information is extremely limited in the manuscript, as the authors only report on some papers, especially regarding the combined use of sorafenib and metformin without giving any details on the mechanism of death:
-See line 326 where they mention a study on xenografts (ref 109)
-Lines 421-423 where they just report a number of works on in vivo models. What is the impact of the combinatorial use of sorafenib and metformin on the modality of cell death induced is not discussed at all.
Besides metformin and VPA there is one other combination (resminostat + sorafenib) that the authors report, but I don't understand the relevance of this combination to the subject of this review, which is the mechanistic findings of cell death…
There is no information on clinical trials according to what the authors report, as they only report on one trial with no results yet. Moreover I may be mistaken but in diabetic patients receiving both sorafenib and metformin the outcome is worse (lines 426-431)? This is a bit unexpected according to what the authors state and they should have provided more details and discussion on this work as it doesn't seem to support the hypothesis of a combination of these two drugs to treat liver cancer. The clinical implementation part is rather unclear under this prism.
In lines 432-434 the authors report “As shown above, multiple evidence exists indicating the induction of pro-apoptotic profiles and cell viability reduction after the treatment with sorafenib alone or in combination with VPA and/or metformin in a dose-dependent manner (Figures 5 and 6)” but this is not so as the authors just report the conclusions of some works without saying anything on the cell death profiles (but in one case, lines 417-419).
Even the section on the mechanisms of cell death related to sorafenib is very poor. The authors failed to report any information on ferroptosis which seems to be a very important cell death modality induced by sorafenib and related to resistance against this drug.
Finally, the concluding part "Future directions and challenges" is poor and rather arbitrary, not clearly supported by what the authors have presented and discussed in the rest of the review.
Author Response
Comment:
The manuscript by Edgar X. Franco-Juárez et al entitled “Mechanistic findings of cell death in drug combinations with sorafenib against liver cancer: from basic research to clinical implementation” is a review manuscript in which authors attempt to provide information on the potential combination of this b-raf inhibitor and other drugs to treat liver cancer. Despite the fact that the subject matter is interesting, this reviewer finds it very difficult to understand the take-home message of this review:
Response:
Thank you for your observation. The style of the manuscript has been improved and some figures have modified accordingly to summarize cell death features by sorafenib, valproic acid and metformin, including potential regulatory proteins shared among RCD subroutines triggered by these drugs.
Comment:
The drug combinations implied by the authors in the title are in fact combinations with VPA and metformin only. However, even this information is extremely limited in the manuscript, as the authors only report on some papers, especially regarding the combined use of sorafenib and metformin without giving any details on the mechanism of death:
Response:
Thank you for your comments. We have made adjustment into our title and subtitles and added information on ferroptosis, pyroptosis and necrosis. We also made a more comprehensive description of the studies and landscape along all the manuscript.
Comment:
-See line 326 where they mention a study on xenografts (ref 109).
Response:
Broader information about xenograft studies has been added.
Comment:
-Lines 421-423 where they just report a number of works on in vivo models. What is the impact of the combinatorial use of sorafenib and metformin on the modality of cell death induced is not discussed at all.
Response:
We have included more information about the impact of combinatorial use of sorafenib and metformin and is also discussed.
Comment:
Besides metformin and VPA there is one other combination (resminostat + sorafenib) that the authors report, but I don't understand the relevance of this combination to the subject of this review, which is the mechanistic findings of cell death…
Response:
Thank you for your observation. Information on resminostat + sorafenib has been removed.
Comment:
There is no information on clinical trials according to what the authors report, as they only report on one trial with no results yet. Moreover I may be mistaken but in diabetic patients receiving both sorafenib and metformin the outcome is worse (lines 426-431)? This is a bit unexpected according to what the authors state and they should have provided more details and discussion on this work as it doesn't seem to support the hypothesis of a combination of these two drugs to treat liver cancer. The clinical implementation part is rather unclear under this prism.
Response:
We have added information about the limitations of drug combinations emphasizing the outcomes of this combination in diabetic patients.
Comment:
In lines 432-434 the authors report “As shown above, multiple evidence exists indicating the induction of pro-apoptotic profiles and cell viability reduction after the treatment with sorafenib alone or in combination with VPA and/or metformin in a dose-dependent manner (Figures 5 and 6)” but this is not so as the authors just report the conclusions of some works without saying anything on the cell death profiles (but in one case, lines 417-419).
Response:
Thank you for your observation. We have added information about cell death mechanisms when using sorafenib, valproic acid and metformin.
Comment:
Even the section on the mechanisms of cell death related to sorafenib is very poor. The authors failed to report any information on ferroptosis which seems to be a very important cell death modality induced by sorafenib and related to resistance against this drug.
Response:
Thank you for the advice. We have added information regarding sorafenib-cell death mechanisms known to be facilitated by valproic acid or metformin.
Comment:
Finally, the concluding part "Future directions and challenges" is poor and rather arbitrary, not clearly supported by what the authors have presented and discussed in the rest of the review.
Response:
Thank you for your suggestion. We have made changes in the section and propose the evaluation of dose-dependancy (as different dose-response have been observed in the single or combinated treatment with sorafenib/VPA or sorafenib /metformin) and limitations to its use (as negative effects has been observed to be related, such as metformin administration followed by sorafenib) please refer to references 154-155.
Reviewer 2 Report
Comments and Suggestions for Authors
Dear Authors,
I would like to extend my congratulations for the strength of your manuscript, particularly in two key areas. First, your focus on the theme of drug repurposing in HCC therapeutics is both interesting and innovative. It adds an exciting dimension to your research and has the potential to have a significant impact in the field. Secondly, your commitment to including recent and up-to-date references is commendable. This not only enhances the credibility and relevance of your work but also demonstrates your dedication to providing the most current information to your readers. These two strengths make your manuscript a valuable contribution to the field.
While I appreciate the strengths and innovative aspects of your manuscript, it's important to address and acknowledge the limitations as well:
· The text lacks a clear and consistent paragraph architecture, making it difficult to follow the flow of ideas and information. I would recommend organizing the content into paragraphs to improve readability and comprehension. Please consider revising the document to ensure that the text is divided into logical and coherent paragraphs.
· I would like to suggest that you expand on the discussion of cell death mechanisms beyond autophagy and apoptosis. These mechanisms, such as necrosis, necroptosis, pyroptosis, ferroptosis, paraptosis, anoikis, and entosis, play critical roles in HCC carcinogenesis. Including a comprehensive overview of these mechanisms in your manuscript would enhance its depth and comprehensiveness. For each of these mechanisms, consider providing information on their basic mechanisms, regulatory pathways, and differences from apoptosis and autophagy. This will give readers a more comprehensive understanding of the various ways in which cells can undergo programmed or pathological cell death.
So either change the title of the manuscript, or analyze in more depth since there is tons of evidence regarding the effects of sorafenib in ferroptosis ( https://pubmed.ncbi.nlm.nih.gov/?term=sorafenib+hcc+ferroptosis&sort=pubdate ), pyroptosis ( https://pubmed.ncbi.nlm.nih.gov/?term=hcc+sorafenib+pyroptosis&sort=pubdate ) and necrosis ( https://pubmed.ncbi.nlm.nih.gov/?term=sorafenib+hcc+necrosis+&sort=pubdate ).
· “Patients within the 0, A and B stages can 67 receive surgical treatment like ablation, liver transplantation, surgical resection, and chemoembolization (TACE) and they have a life expectancy from 2.5 to > 5 years”. It's important to clarify that ablation and TACE are not surgical procedures. Ablation typically involves using heat or cold to destroy tissue, and TACE is an interventional radiology procedure, not a surgical one.
· There is a relevant study on the synergy of sorafenib and metformin that has been published recently, and I recommend citing it in your manuscript to strengthen the support for your research findings ( https://pubmed.ncbi.nlm.nih.gov/37370771/ ). Including this recent study in your references will provide readers with the most up-to-date information and a more comprehensive understanding of the topic.
Addressing limitations can help you provide a more balanced and comprehensive perspective on your findings. It allows readers to better understand the context and boundaries of your research, ultimately contributing to a more robust and well-rounded paper.
Comments on the Quality of English LanguageThe manuscript is generally well-written in English with only minor readability issues. A few minor language and grammar improvements would enhance its clarity.
Author Response
Dear Authors,
I would like to extend my congratulations for the strength of your manuscript, particularly in two key areas. First, your focus on the theme of drug repurposing in HCC therapeutics is both interesting and innovative. It adds an exciting dimension to your research and has the potential to have a significant impact in the field. Secondly, your commitment to including recent and up-to-date references is commendable. This not only enhances the credibility and relevance of your work but also demonstrates your dedication to providing the most current information to your readers. These two strengths make your manuscript a valuable contribution to the field.
Response:
We would like to thank you for your comments. We have made emphasis on these two elements in our manuscript.
Comment:
While I appreciate the strengths and innovative aspects of your manuscript, it's important to address and acknowledge the limitations as well:
Response:
Thank you very much, we are open and looking forward to improve our writing.
Comment:
The text lacks a clear and consistent paragraph architecture, making it difficult to follow the flow of ideas and information. I would recommend organizing the content into paragraphs to improve readability and comprehension. Please consider revising the document to ensure that the text is divided into logical and coherent paragraphs.
Response:
Thank you for your recommendation. Major changes in redaction and in order of some sub-themes have been made it in a logical sequence initiated by individual cell death subroutines triggered by sorafenib, followed by sorafenib-cell death subroutines facilitated by VPA or metformin.
Comment:
I would like to suggest that you expand on the discussion of cell death mechanisms beyond autophagy and apoptosis. These mechanisms, such as necrosis, necroptosis, pyroptosis, ferroptosis, paraptosis, anoikis, and entosis, play critical roles in HCC carcinogenesis. Including a comprehensive overview of these mechanisms in your manuscript would enhance its depth and comprehensiveness. For each of these mechanisms, consider providing information on their basic mechanisms, regulatory pathways, and differences from apoptosis and autophagy. This will give readers a more comprehensive understanding of the various ways in which cells can undergo programmed or pathological cell death.
Response:
Thank you for your valuable suggestion. We have follow-it and add information as seen on section: Sorafenib has the potential to modulate another cell death subroutines.
Comment:
So either change the title of the manuscript, or analyze in more depth since there is tons of evidence regarding the effects of sorafenib in ferroptosis ( https://pubmed.ncbi.nlm.nih.gov/?term=sorafenib+hcc+ferroptosis&sort=pubdate ), pyroptosis ( https://pubmed.ncbi.nlm.nih.gov/?term=hcc+sorafenib+pyroptosis&sort=pubdate ) and necrosis ( https://pubmed.ncbi.nlm.nih.gov/?term=sorafenib+hcc+necrosis+&sort=pubdate ).
Response:
Thank you for your observations. We have made changes in title that help to clarify the objective and extend of this manuscript. We have add a comprehensive description in all cell death subroutines shared between sorafenib, valproic acid and metformin. Cell death subroutines like anoikis, entosis, and others are out of scope, as there is no information available that helps to clarify if valproic acid or metformin could cooperate with sorafenib.
Comment:
“Patients within the 0, A and B stages can 67 receive surgical treatment like ablation, liver transplantation, surgical resection, and chemoembolization (TACE) and they have a life expectancy from 2.5 to > 5 years”. It's important to clarify that ablation and TACE are not surgical procedures. Ablation typically involves using heat or cold to destroy tissue, and TACE is an interventional radiology procedure, not a surgical one.
Response:
Thank you for your recommendation. We have modified our terms.
Comment:
There is a relevant study on the synergy of sorafenib and metformin that has been published recently, and I recommend citing it in your manuscript to strengthen the support for your research findings ( https://pubmed.ncbi.nlm.nih.gov/37370771/ ). Including this recent study in your references will provide readers with the most up-to-date information and a more comprehensive understanding of the topic.
Response:
Thank you for your suggestion. We have read and include the suggested article (see reference 134) along with information obtained from it (see references 133, 143).
Comment:
Addressing limitations can help you provide a more balanced and comprehensive perspective on your findings. It allows readers to better understand the context and boundaries of your research, ultimately contributing to a more robust and well-rounded paper.
Response:
Thank you. We have identified and include a subtitle that state our major limitations, such as lack of studies towards RCD in combinatorial approaches, potential source of bias related to sample size in clinical studies and information retrieval sources.
Round 2
Reviewer 2 Report
Comments and Suggestions for Authors
Thank you for addressing all my comments and enhancing the manuscript significantly. Incorporating aspects from the article "The Role of the NLRP3 Inflammasome in HCC Carcinogenesis and Treatment: Harnessing Innate Immunity" (Cancers, 2022 Jul; 14(13): 3150) PMID: 35804922 is a great move and could fortify the discussion substantially.
Author Response
We thank the reviewer comment. We have revised the manuscript suggested, but it has a little bit complicated to include it in our review.